# Romantic Duration, Relationship Quality, and Attachment Insecurity among Dating Couples

**DOI:** 10.3390/ijerph20010856

**Published:** 2023-01-03

**Authors:** Harry Freeman, Jeffrey Simons, Nicholas F. Benson

**Affiliations:** 1Human Development and Educational Psychology, University of South Dakota, Vermillion, SD 57069, USA; 2Department of Psychology, University of South Dakota, Vermillion, SD 57069, USA; 3Department of Educational Psychology, Baylor University, Waco, TX 76798, USA

**Keywords:** dating relationships, attachment insecurity, romantic duration, relationship quality, romantic relationship trajectories

## Abstract

For many young adults today dating is not taken as a path to marriage, but as a relationship to be considered on its own terms with a beginning, middle, and end. Yet, research has not kept pace as most studies that look at relationships over time focus on marriages. In the present study, we look at individual differences and normative patterns of dating relationship quality over time. We tested a path model of associations between relationship duration, attachment insecurity, and four relationship quality domains (sexual frequency, commitment, satisfaction, and companionship) among a large sample of dating young adults (*N* = 1345). Based on a conceptual model of romantic relationship development, results supported expectations that dating trajectories are curvilinear, with unique patterns of accent, peak, and decent for each relationship domain. Dating duration also moderated the relationship between dating quality and attachment insecurity with anxious attachment becoming a more salient predictor of lower satisfaction and lower commitment in long-term versus short-term relationships. A quadratic interaction with sexual frequency indicated that insecurity predicted less sexual activity in new relationships, more activity among relationships between two and four years, but then less again in longer-term relationships. Findings suggest patterns of stability and change in dating relationships during emerging adulthood that complement those observed from the marriage literature.

## 1. Introduction

Oscar Wilde’s quote, “One should always be in love. That is the reason one should never marry.” [1] has mostly held up to scientific scrutiny. Studies find that marriages, on average, decline in relationship satisfaction, commitment, and sexual frequency over time [2,3,4,5]. Less understood is the corollary of Wilde’s claim, that staying unmarried will make a relationship immune to the negative effects of time, which is the subject of the current investigation. Namely, does time exert a different effect on relationship quality among dating couples? In contrast to the large body of research on marriages, the extant body of evidence on how duration affects dating quality is sparse, inconsistent, and represented by a handful of older longitudinal studies with small samples [6,7,8,9].

In the current study, we seek to fill the gap in understanding how (or whether) dating duration predicts dating quality as indicated by three widely-studied trends in marital quality: relationship satisfaction, dedication commitment, and sexual activity. We also examine whether relationship duration predicts behavioral companionship, a contemporary but less studied relationship factor. Furthermore, because personality factors, such as attachment style, become increasingly salient predictors of marital quality over time [10], we examine if duration moderates the relationship between insecurity and dating relationship quality.

Surprisingly little is known about the quality of dating relationships at different time points. This knowledge gap is likely due to traditional views of dating relationships as a transitory experience in the service of marriage (i.e., courtship) rather than as an end in themselves. Contemporary research has focused more on casual dating scripts among young adults [11,12,13]. Yet, even within the recent context of “hook-up culture”, four in ten college students report being in a “serious romantic relationship” with an average duration of more than 18 months [10,14,15,16,17]. For most college students, especially women, casual scripts are initially viewed as a precursor to dating [18] and the incidence of hook ups significantly diminishes past the first year of college as students pair off into more exclusive dating relationships [6].

Over the past three decades, and increasingly in today’s context of emerging adulthood, stable nonmarried romantic relationships occupy a normative—if not signature feature—of this developmental period [14,19]. Experiences in committed romantic relationships are considered one of the four signature transitions affecting personality development during emerging adulthood [15]. Given the scarcity of research on the associations between dating duration and quality in young adults’ dating relationships, an important next step is to better understand normative and individual trends in dating relationships and determine whether they follow a similar trajectory to their married counterparts. We employ a conceptual model on romantic relationship trajectories [4] to predict differences in dating relationships by duration and attachment orientation.

Eastwick and colleagues [4] proposed the Relationship Trajectories Framework as a conceptual model for evaluating the normative arc in the lifecycle or romantic relationships. According to the framework, the time course of relational qualities can be segmented into three periods including an initial assent, followed by a peak, and an eventual descent. In the current study we examine the normative features of each period (i.e., time, length, and slope of ascent, and timing and stability of peak, and the timing and slope of decent) for relationship satisfaction, relationship commitment, companionship, and sexual activity. In additional to exploring normative trends in these relationship quality domains, we also explore individual differences by attachment style. We first briefly review the voluminous work on marriage quality trends, then present the available work on dating relationships and use social exchange theory to predict why dating patterns may differ from marital relationships.

### 1.1. Normative Trends in Marital Relationships

A half century of marriage research has established relationship duration as a robust and consistent predictor of declining relationship satisfaction, commitment, and sexual activity [17,20,21]. In a meta-analysis, Mitnick and colleagues [5] found that the average newlywed experiences stable to slightly declining levels of relationship satisfaction over the first 18 months of the marriage, followed by a slow to moderate decline that extends into middle adulthood.

Closely linked to relationship satisfaction is the concept of dedication commitment [22] defined as a person’s intent to stay in the relationship and their confidence that the relationship will last into the future [23]. Like satisfaction, dedication commitment declines gradually over the first years of married relationships [3]. Lastly, a number of studies have shown marital duration to be inversely related to sexual frequency, with the steepest declines occurring after the first year—even when controlling for participant age [2,24,25,26,27,28].

### 1.2. Normative Trends in Dating Relationships

The work on stability and change in dating relationships is sparse, inconsistent and old. Sprecher [9] reported that relationship satisfaction and commitment are relatively stable over brief measurement intervals (i.e., less than six months), and increase slightly when tracked over a two to four year period. Rusbult [7] followed 17 young couples in their first months of dating and found significant increases in both satisfaction and commitment nine months later. In contrast to Rusbult’s findings, a somewhat more recent study by Byers [6] followed both dating and married long-term couples and found relationship satisfaction and sexual satisfaction significantly declined over an 18-month period, independent of relationship type.

Studies linking relationship duration to sexual frequency among nonmarried young adults have rarely focused on dating-only samples. In fact, only a single cross-sectional study is available: a study based on 1983 data from the National Survey of Unmarried Women [29]. Tanfer and Cubbins reported an immediate and steady inverse correlation between duration and sexual frequency; specifically, females in relationships of less than six months duration reported 64% more sexual activity compared to those in relationships of at least two years (*M* = 9.0/month versus *M* = 5.5/month, respectively).

In addition to relationship satisfaction, commitment, and sexual frequency, we examine a less frequently studied outcome, the amount of time romantic partners spend together (i.e., companionship). Behavioral companionship is central to notions of romantic unions [30] and can be broadly defined as the amount of awake-time couples spend together. In a recent national survey by the Pew Research Center [31], young adults rated “spending time together” as one of their three most likely reasons to marry and ranked this quality as more important than having children or financial stability. Although important, empirical work on companionship is sparse, quite old, and limited to married samples [32]. It remains unstudied whether relationship duration is related to how much time couples spend together.

### 1.3. Why Dating and Marital Trends May Be Different

Social exchange theory [33] provides a conceptual framework for understanding why dating relationship trends may be different from marriages. Fundamental to the theory is the idea that relationships persist when reasons for staying together exceed those for leaving [34]. Reasons for staying together may include either compelling positive reasons (aspects of pleasure and satisfaction such as sexual interest and emotional fulfillment) and constraints (such as children, shared finances, and mutual social connections) that make it difficult to easily dissolve the relationship. Growing constraints in the face of waning gratification translate to dissatisfied but stable relationships. Relatively new dating relationships are likely to benefit from both sides of the social exchange equation, having higher pleasure and less constraints compared to their married counterparts.

The first months and years of dating relationships are distinct from new marriages because this is the only point of time in the lifecycle of a romance in which the romantic partner is actually novel. Studies show that partner novelty is closely linked to physical attraction, relationship satisfaction, and passion [35,36,37]. Furthermore, couples have more to offer to the relationship when their lives, interests, and personalities are new to each other [38]. Constraints accumulate as couples invest more time, emotion, and resources into a shared life; these constraints consistently predict increased problem-focused interactions and declining relationship satisfaction, with the most dramatic downturn in marital satisfaction following the birth of a child [3,39,40]. Dating couples are generally less likely to be bound by shared finances, children, or social obligations, at least in the short term.

Because early dating is relatively free of relationship constraints and high in partner novelty, it is an ideal environment to spend more time together, have more sex, and grow relationship satisfaction and commitment. In this way, dating relationships should evidence a longer accent than their married counterparts. However, relationship constraints would likely accumulate as the couple’s social worlds become increasingly interconnected [23]. The result, we hypothesize, is that time in a dating relationship will predict a curvilinear trajectory in the four relationship qualities measured.

### 1.4. Individual Differences by Attachment Style

Insecure attachment representations (classified under dimensions of avoidance and anxiousness) consistently predict poor romantic relationship outcomes in both dating and married samples [41]. Highly anxious attachment styles in adults are characterized by distrust of partner loyalties, fear of rejection, and a cognitive preoccupation and obsession with the relationship. In turn, this cognitive set predicts hyperactivation of the attachment system, characterized by bouts of anger, intrusiveness, jealousy, passion, and mania [41,42]. Insecure avoidance is characterized by cognitions and behaviors that dismiss the importance of the attachment relationship and promote emotional distance from partners. Not surprisingly, both of these insecure attachment styles consistently and robustly predict lower relationship satisfaction [42,43,44] and poorer quality sexual experiences [27,45,46,47]. Paradoxically, some studies have found positive associations between anxious attachment and sexual frequency [48,49]: an outcome that is attributed to sexual behavior being used either as a reparative mechanism to diffuse conflict or as a strategy to pursue unmet attachment needs [50].

The effects of insecurity on relationship quality appear to become stronger as relationships age. In a meta-analysis of 57 cross-sectional studies including both dating and married samples [38] indicated that the negative association between attachment insecurity and relationship satisfaction was stronger in study samples with longer mean relationship durations. This effect was similar for both insecure anxious and insecure avoidant dimensions of attachment insecurity. The authors suggest that the novelty and flood of positive emotions in new relationships mask personality differences and overshadow negative events but as novelty diminishes, the deleterious effects of insecurity are more likely to be felt. Consistent with Hadden and colleagues’ temporal model, we hypothesize that relationship duration will moderate the association between attachment insecurity and relationship quality, marked by increasingly negative associations between an attachment styles and each of the four relationship quality dimensions.

## 2. Method

### 2.1. Recruitment

All study participants were undergraduate students at a public Midwestern University in the United States. A total of 2487 participants completed a 20 min online survey on dating relationships. The data were collected over a period of six semesters between fall 2007 and spring 2013. Participants completed the survey either as part of a required assignment for research participation or to get extra credit for the class.

### 2.2. Sample

The study was open to all undergraduate students; however, because the focus of the study was on traditional undergraduate students in romantic relationships, the statistical analysis excluded students who were 30 years of age or older (n = 49), were married (n = 5), or were not in a serious dating relationship (n = 1033). In addition, 14 surveys were dropped because an examination of patterned and inconsistent responses indicated the answers were not sincere, and 41 surveys were dropped because the survey was not completed in a reasonable timeframe (approximately 20 min). A reasonable time for survey completion was piloted with 20 undergraduate students. Unreasonable completion times are defined as times that exceeded 2 *SDs* less than the piloted time.

The final study sample comprised 1345 young adults (72% female) in dating, relationships. All participants were students at a public Midwestern University in the United States. Participants ranged in age from 18 to 23 years; however, the majority of participants were freshmen (45%) and sophomores (24%) and were under 20 years of age (*M* = 19.7, *SD* = 1.77).

### 2.3. Measures

Demographics. Participants provided information on their gender, age, romantic status, romantic relationship duration, living situation, and socioeconomic status (income and parents’ education). Socioeconomic status (SES) was computed as a continuous measure by summing two 4-point scales, including parents’ income (1: <20,000; 2: 20,000–40,000; 3: 40,000–80,000; 4: >80,000), and highest education background attained by at least one biological or step-parent (1: High school; 2: Some college; 3: College degree; 4: Professional or graduate degree). About one in five participants (20.9%) reported that their biological parents were either divorced or separated. Ethnicity was not measured due to the largely homogenous university population (>93% non-Hispanic White) from which the sample was drawn.

Romantic status was assessed with a single item, “Do you currently have a boy/girlfriend, or are you seriously dating someone?” Response options included “Not seriously dating”, “Seriously dating”, “Engaged”, and “Married”. Only students in seriously dating relationships were included in the sample. A question on “living situation” was used to classify participants into cohabitating relationships; of those in dating relationships, 34 (2.5%) were living with their romantic partner. When controlling for age and romantic duration, cohabitators did not differ significantly from non-cohabitators on attachment insecurity, sexual frequency, relationship satisfaction, or relationship commitment, albeit not surprisingly cohabitators reported considerably higher companionship (*M* = 30.35 h per week versus 17.84 h, respectively; *F*(1344) = 58.45, *p* < 0.0001). Given the small sample of cohabitating participants and given the lack of differences on all but one study variable, and based on work indicating that this relationship status shares much in common with dating relationships [51], the decision was made to include these participants in the final sample for the path analysis.

### 2.4. Romantic Relationship Duration

Relationship duration was assessed for those in dating relationships with the following forced-choice question: “How long have you been dating this person?” Response categories and the percentage of responses in each category are as follows: Less than one month (8%); 1–2 months (11%); 3–5 months (12%); 5–8 months (11%); 9–12 months (9%); 1–2 years (23%); 2–3 years (12%); 3–4 years (8%); and More than 4 years (6%). Duration categories were converted to months by taking the average of each interval (e.g., 1–2 years = 18 months). The mean romantic relationship length was 17.2 months. Just under half the participants in the sample reported being in relationships of less than 1-year duration.

Relationship commitment. A latent variable was created to measure dedication commitment, as defined by Rhoades et al. [23]. Drawing on Little et al. [52] item-test correlations method, three parcels were developed using five items from existing scales that reflect confidence in longevity of relationship and intent to stay in the relationship [53,54]. Sample items for relationship confidence include “How sure are you that this relationship will last no matter what?” and “How sure are you that this person will continue to be an important part of your life in the future?” Sample items for intent to stay in the relationship include “I want to spend my life with him or her” and “I will always be loyal to her/him”. All items were rated on 5-point scales.

Relationship satisfaction. A latent variable was created using three items from existing scales, including two items from the relationship satisfaction subscale of the Network of Relationships Inventories (NRI) [53]: “How satisfied are you with your relationship to this person?” and “How happy are you with the way things are between you and this person?” Both items were scored on a 5-point Likert scale. A third item came from the Love Style Inventory (LSI) [54]: Participants were asked how much they agree with the statement “This relationship has met my best expectations”.

Companionship. Behavioral companionship was based on how many waking hours per day participants spent with their romantic partner during an average week. Two separate items were developed to measure the amount of time spent together during weekdays and weekends as follows: “How many hours during the day, on average, do you spend with [your boy/girlfriend] during the [week/weekend]?” Response options for the weekday item included: (1) Less than 1 h per day; (2) 1–2 h per day; (3) 2–3 h per day; (4) 3–5 h per day; and (5) More than five hours per day. The second item, which assessed weekend hours, provided an additional option, “More than 10 h per day”. Responses were converted to hours and summed across the two items (see Table 1 for *M* and *SD*).

Sexual frequency. Participants were asked how many times in the past month they had sexual intercourse with their current romantic partner. Seven response options were provided: (1) Almost every day; (2) More than three times per week; (3) Two to three times per week; (4) About once per week; (5) Two to three times per month; (6) About once per month or less; and (7) I have not had sexual intercourse with my current romantic partner. Responses were reverse coded and converted to measure frequency per month (see Table 1 for *M* and *SD*).

Attachment insecurity. A latent variable, attachment insecurity, was based on six items from the anxiousness subscale of the Experience in Close Relationships—Revised (ECR-R) [55]. In order to reduce redundancies (and consequently measurement error) and to simplify interpretation, three parcels based on item correlations were created to load on a single latent variable. Although subjects completed items from the ECR avoidance subscale, this dimension was not included in the analysis due to a poor fit with the measurement model (see Section 3.2).

## 3. Results

### 3.1. Descriptive Statistics

Table 1 displays the descriptive statistics for observed variables. As indicated in the table, data screening revealed no atypical skew or kurtosis. The total percent of missing data for the study was 3 percent.

Intercorrelations among the observed variables are displayed in Table 2. The distribution of the two behavioral outcomes, sexual frequency and companionship, are worth noting. The average sexual frequency (*M* = 7.17 per month) is consistent with Tanfer and Cubbins [29] report from the 1980’s (*M* = 6.70), as well as with more recent work with U.S. samples of similar age [17].

Additionally, noteworthy is the bimodal distribution in the current sample: 48% of participants reported an average coital frequency of over 10 times per month, and another third of the sample reported either abstinence (20%) or a coital frequency of less than once per month (14%). Abstinence was relatively common for new relationships only. Approximately one in three individuals reported no sexual activity in relationships of less than six months duration, whereas this frequency dropped to one in ten by the half-year mark, and did not change significantly beyond this point.

In terms of companionship, young adults, on average, spent nearly three hours per day with their romantic partner; however, this figure included a significant minority of participants (13%) who spent little (less than four hours per week) or no time with their partners. Not surprisingly, 76% of this group reported little or no sexual activity in their relationship, possibly due to being in long-distance relationships. A third of the participants in the sample reported spending between 18 and 24 h per week with their romantic partner and another 13% of individuals fell into the highest category, spending more than 35 h per week with their partner.

### 3.2. Measurement Model

The measurement model included three latent variables (commitment, insecurity, and relationship satisfaction) with three indicators each. After model testing, the insecurity latent factor was limited to the insecure anxious items from the ECR. The model including attachment avoidance defined by three avoidant items from the ECR, was an adequate fit to the data χ^2^ (48, *N* = 1345) = 451.522, *p* < 0.001, CFI = 0.96, RMSEA = 0.079 90% CI [0.072, 0.086], SRMR = 0.030. However, modification indices suggested potential cross loadings of an avoidance item (not showing deep feelings) on the insecurity factor as well as a cross-loading of a relationship satisfaction indicator (LSI expectation) on the avoidance factor. In addition, one of the avoidance items had a somewhat low standardized loading of 0.57. The model with anxious attachment items and not avoidance items had better fit indices and a clearer factor structure.

The model was estimated in Mplus version 8.4 [56] with the weighted least squares means and variances (WLSMV) estimator to accommodate the ordinal indicators. The measurement model was a good fit to the data, χ^2^ (24, *N* = 1345) = 135.74, *p* < 0.001, CFI = 0.98, RMSEA = 0.059 90% CI [0.049, 0.069], SRMR = 0.020. Factor loadings presented in Table 3 indicate that the observed variables are good indicators of their respective latent variables. Standardized factor loadings ranged from 0.78 to 0.91. Correlations among the latent variables (see Table 3) ranged from −0.33 (commitment with insecure anxious attachment) to 0.74 (satisfaction with commitment).

### 3.3. Structural Models

We first tested a baseline structural model with the WLSMV estimator that did not include hypothesized interactions or quadratic effects. The model included the latent variables from the measurement model and four observed variables; gender, relationship duration, sexual frequency, and companionship. Gender, relationship duration, and insecurity were exogenous variables that predicted four correlated relationship outcomes (relationship satisfaction, relationship commitment, companionship, and sexual frequency). The baseline structural model was a good fit to the data, χ^2^ (48, *N* = 1345) = 271.87, *p* < 0.001, CFI = 0.97, RMSEA = 0.059 90% CI [0.052, 0.066], SRMR = 0.023. Although gender main effects are common in the marriage literature, with males reporting lower commitment and lower relationship satisfaction (Sprecher, 1999 [8]), longitudinal studies reveal similar rates of decline for both husbands and wives in married relationships (Karney & Bradbury, 1995 [34]).

In the second structural model, a quadratic relationship duration effect was added with paths to each of the four relationship outcomes. This model was also a good fit to the data, χ^2^ (54, *N* = 1345) = 272.62, *p* < 0.001, CFI = 0.97, RMSEA = 0.055 90% CI [0.048, 0.061], SRMR = 0.022. The quadratic effect was significantly associated with each of the outcomes except relationship satisfaction.

In the third structural model, we tested the hypothesized effects of the interaction between insecurity and relationship duration on the relationship outcome variables. The initial model included interactions between insecurity and both the linear and squared duration variable. In this model, the insecure interaction with the squared term was only significantly associated with sexual frequency. Hence, the remaining higher order interactions with the squared duration term were dropped for parsimony. In addition, the insecurity by duration interaction was not significantly associated with companionship; hence this was dropped for parsimony, resulting in the final model presented in Figure 1. Traditional fit indices are not available for the model with the latent variable interaction as it requires numerical integration. However, AIC and BIC are provided (AIC = 47,866.50; BIC = 48,256.81). The model accounted for substantial amounts of variance in commitment (*R*^2^ = 0.22, *p* < 0.001) and satisfaction (*R*^2^ = 0.24, *p* < 0.001). However, the model was a relatively poor predictor of frequency of sexual activity (*R*^2^ = 0.02, *p* = 0.014) and companionship (*R*^2^ = 0.03, *p* = 0.001).

Relationship commitment. Results indicate that relationship commitment increases by about 0.44 standard deviations per year at the relationship duration mean (1.43 years) and, on average, commitment peaks 2.99 years into the romantic relationship. Subsequently, relationship commitment begins to decline. However, these associations between relationship duration and commitment are qualified by the significant duration by insecurity interaction (β = −0.09, *p* = 0.003). Individuals who were more anxiously attached exhibited a weaker positive association between duration and commitment. For example, at the sample mean of relationship duration, for individuals high on insecurity (*M* + 1 *SD*) the effect of duration on commitment is *b* = 0.29, *p* < 0.001, whereas for those low in insecurity (*M* − 1 *SD*), the effect is *b* = 0.43, *p* < 0.001. Alternatively, the interaction can be illustrated by the difference in the effect of insecurity at early (*b* = −0.19, *p* < 0.001) versus later (*b* = −0.39, *p* < 0.001) stages of the relationship. Hence, the results are consistent with our hypothesis that insecurity exhibits a stronger deleterious effect on commitment as the relationship progresses. As shown in Figure 2, insecurity is associated with a lower initial commitment to the relationship coupled with a modestly weaker rate of increase over time relative to less anxiously attached participants. Women reported greater relationship commitment than men.

Relationship satisfaction. In contrast to the associations between duration and relationship commitment, duration exhibited little association with relationship satisfaction (see Figure 2). In fact, significant associations between duration and relationship satisfaction were limited to the earlier stages of the relationships for those who were low on insecurity. The interaction between insecurity and duration was significantly associated with satisfaction (β = −0.09, *p* = 0.008). For example, for individuals who were low on insecurity (*M* − 1 *SD*) the effect of duration on satisfaction was *b* = 0.19, *p* = 0.017 at duration *M* − 1 SD and decreased to *b* = 0.0, *p* = 0.003 at the duration mean of 1.43 years. For those with mean or higher levels of insecurity, duration did not have significant effects on relationship satisfaction across the durations ranging from *M* ± 1 *SD*, *p*’s > 0.06. Alternatively, the interaction can be illustrated by the difference in the effect of insecurity on satisfaction at early (*b* = −0.43, *p* < 0.001) versus later (*b* = −0.62, *p* < 0.001) stages of the relationship. Hence, these results are consistent with the hypothesis that insecurity exhibits a stronger deleterious effect on satisfaction as the relationship progresses. On average, relationship satisfaction peaked at a similar time to commitment at 2.98 years. Given the relatively weak direct effects of duration and lack of gender effects, it appears that the substantial effects of insecurity account for most of the explained variance in relationship satisfaction.

Companionship. The associations between relationship duration, insecurity, and companionship are presented in Figure 3. Companionship increases over the course of the first 2.70 years before starting to decrease. At the sample mean for relationship duration, the rate of increase in time spent together is approximately 1.19 h per week for each year of the relationship. The rate of change is roughly double earlier in the relationship (e.g., *M* − 1 *SD*; which corresponds to approximately the first 3 weeks, *b* = 0.25, *p* < 0.001).

Frequency of sexual activity. The results show complex associations between relationship duration, insecurity, and frequency of sexual activity. These are reflected in the significant relationship duration quadratic effect, the insecurity by duration interaction (β = 0.15, *p* < 0.001), and the insecurity by duration^2^ interaction (β = −0.11, *p* = 0.003). For those high in insecurity there were positive associations between relationship duration and frequency of sexual activity at duration *M* − 1 *SD* (*b* = 2.06, *p* < 0.001) and the relationship duration mean (*b* = 0.83, *p* = 0.001). As shown in Figure 3, individuals with greater insecurity showed a greater rate of increase in sexual activity in the early stages of the relationship. In contrast, those low in insecurity exhibited a fairly constant decline in frequency of sexual activity across time. Contrary to the effects of insecurity on commitment and satisfaction, the inverse effect of insecurity on frequency of sexual activity was strongest in the earlier stages of the relationship (*b* = −1.15, *p* = 0.001) and diminished and changed sign for those in longer-term (*M* + 1 *SD*) relationships (*b* = 0.68, *p* = 0.012). On average, frequency of sexual activity was highest approximately 1.78 years into the relationship.

Summary. The hypothesized quadratic trends, marked by short-term positive associations between duration and relationship quality followed by a period of stability and then negative associations, were primarily supported for the relationship quality indicators. For relationship satisfaction, the quadratic trend followed a similar pattern but was not significant. The hypothesized interaction between relationship duration and anxious attachment was significant for three (commitment, satisfaction, and sexual frequency) of the four relationship quality outcomes. For commitment and satisfaction, the form of the interaction was consistent with expectations: anxious attachment had a stronger negative association in longer term relationships. The pattern was different for sexual frequency. Anxious attachment was associated with significantly less sexual activity in short-term relationships, followed by a positive association in longer-term relationships (See Figure 3).

## 4. Discussion

For much of the twentieth century, young adult dating relationships were viewed as a prelude to marriage: defined more as a process of becoming (e.g., courtship) than being an end in and of themselves. Today, Western-educated young men and women spend much of their third decade of life (i.e., 20–29 years of age) in committed non-marital unions, yet there has been limited attention directed at the lifecycle of dating relationships. The current study sheds new light on normative trends and how individual differences in attachment styles predict dating relationship quality at different time points in relationships among emerging adults in college.

The study hypotheses were mostly supported by the findings. The results showed a curvilinear association between dating duration and relationship quality, with each relationship outcome peaking at different time points. The frequency of sexual activity plateaued first, and was highest among participants who reported a relationship duration of between 6 and 12 months. Companionship (measured by the number of hours per week the couple spent together) was the second outcome to plateau, and was highest among participants who indicated a relationship duration of between 1 and 2 years. Finally, commitment was the third outcome to plateau, and was highest among participants who reported relationship durations of between 2 to 3 years.

One possible implication of the curvilinear trends is that young adult dating relationships follow a predictable sequence of relationship quality stages, and off-timing may indicate problematic social and personal adjustment. This life-course approach is popular in describing normative and atypical patterns in adolescent romantic experiences, both in predicting when adolescents form their first romantic partnerships [57] and in predicting the sequence of intimate behavior within dyads [58]. A number of adolescent studies have shown that early starters are more likely to exhibit problematic social and personal behaviors [57]. 

If a sequential model were to be applied to young adult dating relationships, normative and atypical trajectories could be examined both within each quality indicator (e.g., companionship, sexual behavior) and as a progression relative to one another as a function of assent, peak, and descent [4]. For instance, based on the path model observed in the current study, we would expect sexual activity and companionship to peak before attachment and commitment. Individuals who show variations from this pathway may exhibit different patterns of adjustment: an outcome that would be supported by our findings which show sexual activity to peak later than commitment among anxious insecure young adults. The duration by insecurity interactions indicate different patterns of relationship trajectories. More anxiously attached individuals begin relationships less committed, increase commitment at a slower rate, and peak sooner than their less insecure counterparts. In addition, a steeper and sooner descent among more anxious individuals reveals that commitment in the long term has returned to the level of new relationships. In contrast, less insecure individuals in long term relationships report considerably higher commitment levels than their counterparts in new relationships. A similar duration by insecurity interaction is observed for relationship satisfaction. These findings are consistent with Hadden and colleagues’ [38] Temporal Adult Romantic Attachment Model (TARA) which proposes that the negative effect of insecurity on relationship quality increases as relationships age.

Contrary to expectations, sexual frequency was not directly related to insecure attachment and duration predicted a linear decline with increasing duration. Yet, a duration by insecurity quadratic interaction for sexual activity revealed different relationship trajectories. Higher levels of insecurity predicted an curvilinear trend marked by increasingly higher levels of sexual activity among longer term couples with a peak between two and three years, and then diminishing levels at a similar rate as the ascent. In contrast, relationship duration among less insecure individuals predicted a steady linear decline in sexual frequency that is more moderate than the slope of the ascent or descent among more anxiously attached individuals. Higher scores on anxious attachment predicted less sex in new relationships, more sex in relationships between two and three years, and similar rates of sexual activity in relationships beyond three years.

The duration by insecurity interaction pattern is complex and does not conform to a straightforward interpretation according to the TARA model, nor does the normative trajectory follow an arc pattern as predicted by Eastwick et al. [4] conceptual framework. Nonetheless, the findings are consistent with the only previous study [29] providing duration by sexual activity data among dating couples, which was also cross-sectional. Unlike the other relationship quality factors explored here—especially relationship satisfaction and commitment—the relative importance of sexual activity changes with relationship duration.

In reviewing the consistent declines in sexual frequency that accompanies the first years of marriage and cohabitation, Schwartz and Young posed the following question: “Is sexual frequency, then, not important for relationships in the long run?” [59] (p. 3). A few recent studies of long-term relationships indicate that sex is not associated with relationship satisfaction beyond a minimum amount or among couples who have sex at least once per week [60,61]. In long-term bonds, the link between sexual frequency and relationship satisfaction appears to operate much like the well-documented effect of income on happiness [46,62]. Having more sex is better to a point, beyond which it is no longer predictive of relationship satisfaction or individual happiness.

For insecure anxious individuals, the sex-satisfaction link may grow stronger as relationships age, at least initially. This hypothesized pathway is consistent with two previously documented findings specific to anxious attachment [63]. First, this group is more likely to perceive conflict as their relationships age [64]. After the honeymoon period of dating relationships, personality factors become more salient, thus leading to higher rate of problem-focused interactions among individuals with insecure anxious attachment styles. Second, anxious individuals are more likely to exploit physical intimacy as a way to repair or ameliorate feelings of distance or rejection. In another study, Barbaro and colleagues [65] cast this strategy in evolutionary terms, showing that anxious attached women and men in long-term relationships (*M* duration = 63 months) exhibited more mate retention behaviors such as sexual inducements and appearance enhancements than their less anxious, more secure counterparts. More recently studies have examined microgenetic processes related to attachment and conflict in daily interactions in romantic relationships [44,66,67]. Over the long term, steady conflict may lead to the abandonment of physical intimacy as a reparative strategy among partners that become increasingly emotionally distant, untrusting, and dissatisfied with their relationship.

The between-subject findings reported in this study can only speak to differences by relationship duration and are unable to confirm that such differences are due to relationship change. The trends observed may be partially attributed to lower quality relationships breaking up sooner, thereby inflating the short-term positive trends and attenuating the negative slope over the long term. Social exchange theory would predict that dissatisfaction in dating relationships with few shared obligations (constraints) will tip the scale toward dissolution rather than staying together. This selection bias, however, is also problematic in longitudinal studies that show higher attrition among less satisfied, low constraint, relationships. In other words, the good news of happier long-term relationships couples is partly predicted by having survived that long, regardless of how the data was gathered.

The between-group design also limited our focus to one relationship trajectory dimension, shape, of the five dimensions presented by Eastwick and colleagues [4]. Within-subject data is necessary to capture trajectory dimensions such as fluctuation and density that elude cross-sectional research. In addition, longitudinal studies that include personality factors such as attachment quality and relationship factors such as communication patterns are likely to reveal individual patterns within these trajectory domains [39,49]. A sequential design would be ideal to isolate relationship trajectories from individual maturation, personality differences, and cohort effects.

The current findings offer new information on the nature and course of twenty-first century dating among emerging adults. A new era of changing economic and social factors contribute to dating as a stand-alone romantic experience, rather than as a temporary way-station to marriage [31]. The continuing trends of extending the time spent in post-secondary education and delayed career employment have led to increased long-term dating during the college years. As nonmarital romantic partnerships grow both in number and in average duration, the forces and experiences that explain the vicissitudes of romantic relationships over time may be less a facet of marital status and more a function of relationship duration.

The current findings support the notion that after a few years, dating relationships begin to follow marital relationship trajectories. For individuals in their initial months and years of dating, a longer relationship means, on average, greater commitment, more sex, and more time spent with romantic partners. Although these initial trends stand in contrast to the normative declines observed in the first years of marriage, the question remains whether these short-term gains are unique to dating or just unique to the first years of new romances. Based on an average period of premarital courtship of nearly three years [68], dating couples who are three to four years into their relationships would be comparable to married couples in their first two years of marriage: a comparison that reveals similar relationship quality trends such as stable relationship satisfaction and declining relationship commitment and sexual activity. In other words, if we start counting relationship duration from the time of a relationship’s inception, duration may exert a similar force on romantic relationship quality independent of relationship status. An important next step is to examine whether the social exchange equation of positive relationship features and relationship constraints predicts long-term dating quality and dating dissolution in a manner that is either similar to or different from the marital literature.

## 5. Limitations 

As previously noted, the cross-sectional design of this study merits caution when interpreting the observed trends in dating relationship quality. Although these are preliminary results, the findings are compelling given the scope and size of this study, currently the largest single study to examine relationship quality trends and personality differences among young adults in dating relationships.

This study is also limited by its reliance on single-source, self-reported data. The use of couple data would add significantly to an understanding of whether differences in self-perceived relationship quality are validated or reciprocated by the partner’s views. Recent works points to dyadic imbalances in relationships that are potent indicators of relationship quality, including socioeconomic inequality [69], and competing goals [70], and personality and attachment style mismatch [71]. Lastly, the current study was not able to investigate attachment avoidance as a our work does not consider background experiences that can help explain individual differences in attachment styles [72]. Unfortunately, the current study dropped the measure of avoidant attachment as a predictor in the final model due to measurement constraints. While anxious and avoidant styles are moderately correlated in most romantic relationship research, the predictive value of each dimension is often distinct [41].

Another limitation worth noting is the relatively homogeneous sample with respect to ethnicity, age, SES, and sexual orientation. The results should be interpreted with caution when generalizing to diverse groups, especially sexual minority populations. Both past and current research has shown that indictors of relationship quality, such as sexual frequency, are perceived differently by gay, lesbian, and heterosexual couples [73,74]. In addition, the sample is restricted to an early adult population who are attending college and non-cohabitating. Relationship constraints and dating expectations are likely to be different among older emerging adults who are living together with full time employment in the community.

## 6. Conclusions 

To summarize, the current study not only provides empirical support normative pathways in dating relationships but also points to the significant role that dating duration plays in moderating links between personality and relationship quality, in which insecurity was shown to exert a stronger effect on longer-term relationships. Overall, these results provide new information on the lifecycle of twenty-first century, college-aged young adult dating relationships. Additionally, the results provide a basis for further examination of these trends both for other populations and for replication in longitudinal studies.

## Figures and Tables

**Figure 1 ijerph-20-00856-f001:**
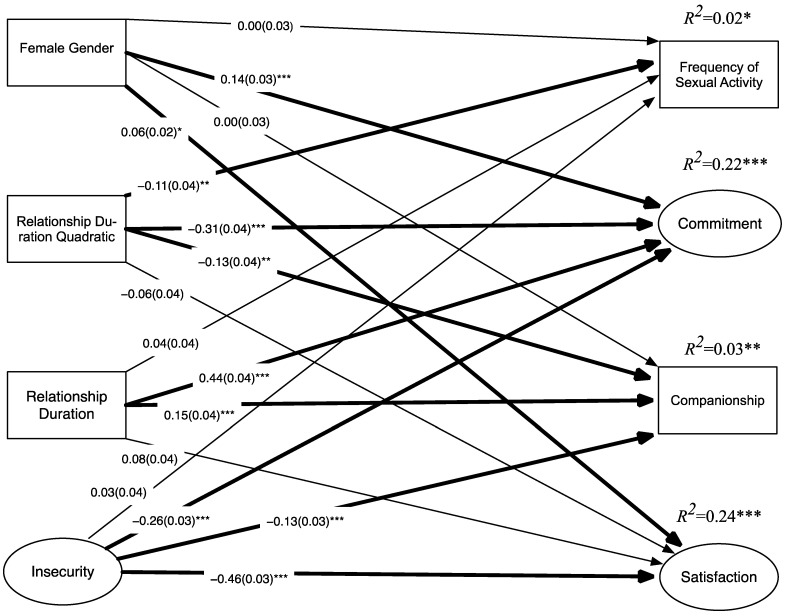
Final model (*N* = 1345). Interactions between insecurity and duration are estimated but omitted from the figure. See text for description of interactions. Coefficients are standardized. *SEs* are in parentheses. Significant effects are in bold. Covariances between the disturbance terms as well as between the exogenous variables are estimated but omitted for clarity. ** p* < 0.05, ** *p* < 0.01, *** *p* < 0.001.

**Figure 2 ijerph-20-00856-f002:**
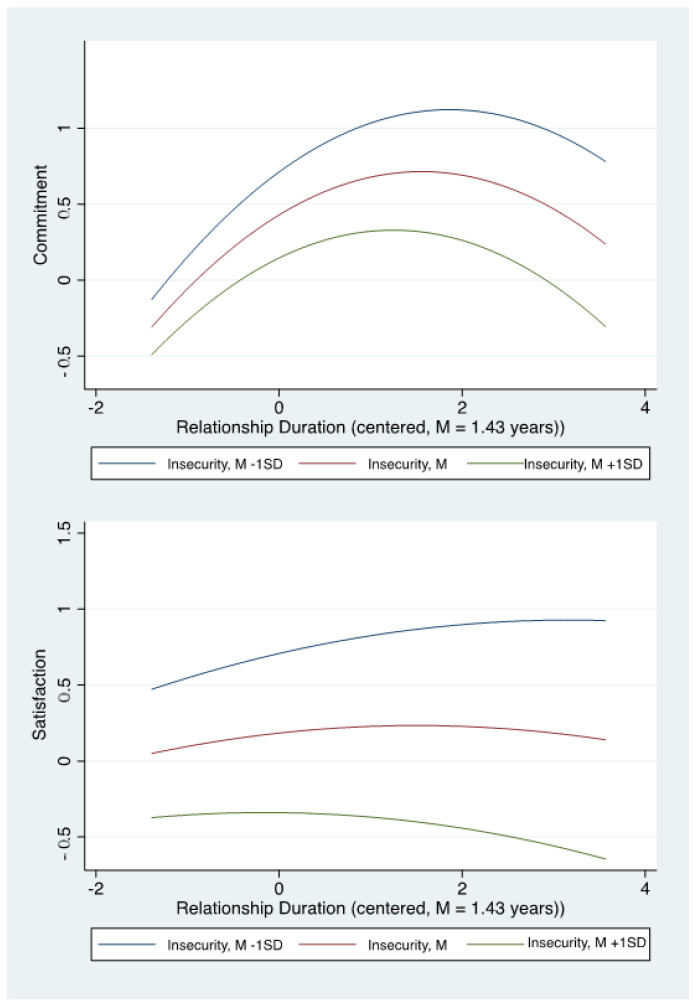
Associations between relationship duration and commitment and satisfaction as a function of insecurity.

**Figure 3 ijerph-20-00856-f003:**
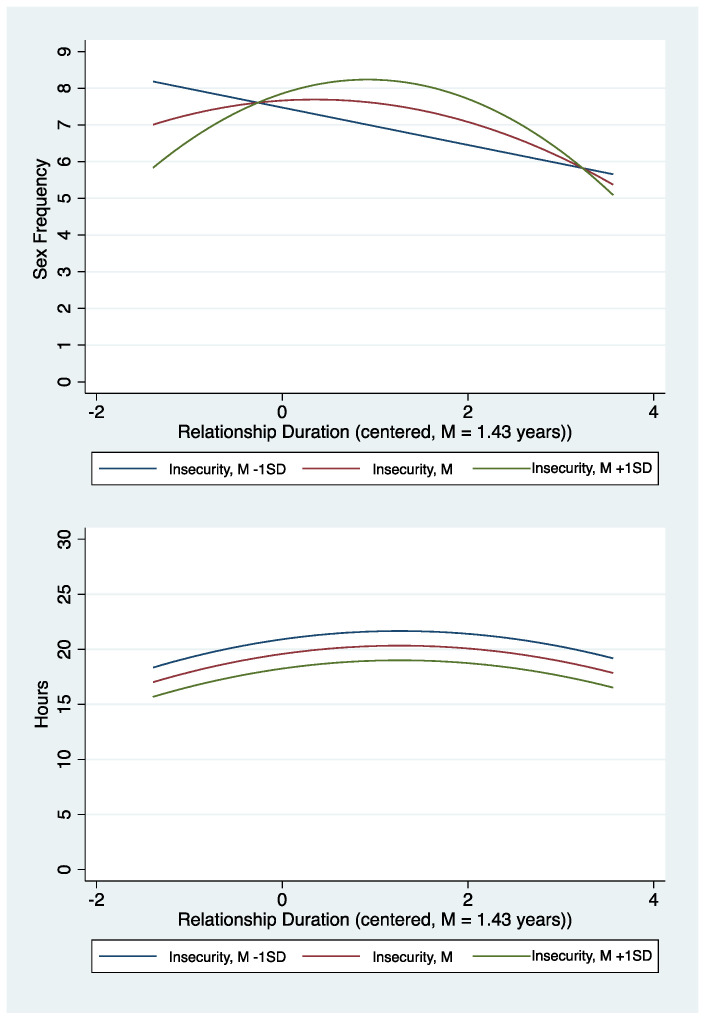
Associations between relationship duration and companionship and sexual frequency as a function of insecurity.

**Table 1 ijerph-20-00856-t001:** Descriptive Statistics for Observed Variables.

Variable	*N*	*M*	*SD*	Min	Max	Skew	Kurtosis
Com	1342	18.67	10.56	2.00	36.00	0.09	2.01
Dur	1347	1.43	1.37	0.04	5.00	1.15	3.54
I1	1338	2.25	0.94	1.00	5.00	0.46	2.62
I2	1338	2.43	1.07	1.00	5.00	0.36	2.25
I3	1338	2.71	1.08	1.00	5.00	0.09	2.13
RC1	1337	3.29	0.81	1.00	4.00	−0.91	3.08
RC2	1342	−0.01	0.85	−3.24	0.92	−1.00	3.47
RC3	1342	−0.00	0.88	−3.07	0.95	−0.78	2.95
RS1	1328	4.06	1.02	1.00	5.00	−0.88	2.94
RS2	1332	4.03	1.07	1.00	5.00	−0.94	3.07
RS3	1338	3.34	0.73	1.00	4.00	−0.93	3.51
SXF	1302	7.17	6.01	0.00	22.00	0.41	2.16

Note: Com = companionship, Dur = relationship duration in months, I1 to I3 = indicators of insecurity, RC1 to RC3 = indicators of relationship commitment, RS1 to RS3 = indicators of relationship satisfaction, SXF = sexual frequency.

**Table 2 ijerph-20-00856-t002:** Correlations among Observed Variables.

	Gender	Com	Dur	I1	I2	I3	RC1	RC2	RC3	RS1	RS2	RS3	SXF
Gender	1												
Com	0.02	1											
Dur	0.07 *	0.08 **	1										
I1	0.11 **	0.12 ***	0.09 **	1									
I2	0.03	0.15 ***	0.19 ***	0.77 ***	1								
I3	0.02	0.12 ***	0.14 ***	0.73 ***	0.77 ***	1							
RC1	0.13 ***	0.20 ***	0.25 ***	−0.20 ***	−0.20 ***	−0.17 ***	1						
RC2	0.17 ***	0.23 ***	0.14 ***	−0.28 ***	−0.25 ***	−0.21 ***	0.69 ***	1					
RC3	0.15 ***	0.22 ***	0.28 ***	−0.26 ***	−0.26 ***	−0.22 ***	0.73 ***	0.75 ***	1				
RS1	0.07 **	0.27 ***	0.12 ***	−0.34 ***	−0.30 ***	−0.28 ***	0.45 ***	0.50 ***	0.52 ***	1			
RS2	0.11 ***	0.22 ***	0.09 **	−0.35 ***	−0.32 ***	−0.30 ***	0.48 ***	0.55 ***	0.57 ***	0.74 ***	1		
RS3	0.07 **	0.21 ***	0.07 **	−0.36 ***	−0.33 ***	−0.30 ***	0.45 ***	0.46 ***	0.48 ***	0.54 ***	0.59 ***	1	
SXF	0	0.31 ***	−0.04	0.01	−0.05	−0.04	0.08 ***	0.05	0.08 **	0.06 *	0.08 **	0.09 **	1

Note: *N*s vary from 1292 to 1344. Com = companionship, Dur = relationship duration in months, I1 to I3 = indicators of insecurity, RC1 to RC3 = indicators of relationship commitment, RS1 to RS3 = indicators of relationship satisfaction, SXF = sexual frequency. * *p* < 0.05, ** *p* < 0.01, *** *p* < 0.001.

**Table 3 ijerph-20-00856-t003:** Standardized Loadings and Latent Variable Correlations for the Measurement Model.

Observed Variable	Insecurity	Relationship Commitment	Relationship Satisfaction
Insecurity 1	0.91	-	-
Insecurity 2	0.87	-	-
Insecurity 3	0.81	-	-
Relationship Commitment 1	-	0.86	-
Relationship Commitment 2	-	0.84	-
Relationship Commitment 3	-	0.88	-
Relationship Satisfaction 1	-	-	0.86
Relationship Satisfaction 2	-	-	0.92
Relationship Satisfaction 3	-	-	0.78
Latent factor			
Insecurity	1.00		
Commitment	−0.33	1.00	
Satisfaction	−0.49	0.74	1.00

Note: *N* = 1345. All factor loadings and correlations are significant at *p* < 0.001.

## Data Availability

The data presented in this study are available on request from the corresponding author. The data are not publicly available.

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
