# Peer review of "Romantic Duration, Relationship Quality, and Attachment Insecurity among Dating Couples"

_ijerph, 2023, doi:10.3390/ijerph20010856_

Round 1

Reviewer 1 Report

Uncommon terminologies could be explained more for the ease of the reader's understanding.

Author Response

Thank you for your positive review.  We have proofed the manuscript for grammatical errors and revised or explained terminology to improve readability.  

Reviewer 2 Report

Introduction

Page 4. I have not understood why, after having mentioned both anxious and avoidant attachment styles throughout the Introduction, suddenly you excluded avoidant attachment style. “A recent meta-analysis of 57 cross-sectional studies including both dating and married samples (Hadden, et al., 2014) indicated that the negative association between insecure attachment, namely anxious attachment style, and relationship satisfaction was stronger in study samples with longer mean relationship durations”. “Consistent with Hadden and colleagues’ temporal model, we hypothesize that relationship duration will moderate the association between anxious attachment and relationship quality, marked by increasingly negative associations between an anxious attachment style and each of the four relationship quality dimensions”. If I remember well, Hadden and colleagues (2014) found the same associations between relationship satisfaction and anxious and avoidant attachment styles. Please, add the Hadden’s results referred to avoidant attachment.

Method

Measures

Attachment insecurity

In my opinion, this part is unclear.

Why did you not include avoidance subscale in the analysis? I mean, why did you not use both subscales of the ECR-R (taken separately), in order to examine the associations between relationship quality and anxious and avoidant attachment styles?

Why did you refer to “attachment insecurity” if you used only six items from the anxiousness subscale of the ECR-R? It is misleading and confusing.

It would be very interesting to evaluate the associations between avoidant attachment style and relationship quality in the sample, the moderation role of relationship duration on these associations, as well as any differences between avoidant and anxious attachment in the model.

Discussion

I think that the study has some major limitations that you should declare and consider in the Discussion section. Firstly, the results are not generalizable to emerging adulthood (18-29 years) because participants were undergraduate students and ranged in age from 18 to 23 years (M=19.7, SD=1.77). They were very young emerging adults involved in romantic relationships, in fact, only the 2.5% of them were living with their romantic partner. Although you observed that “cohabitators did not differ significantly from non-cohabitators on attachment insecurity, sexual frequency, relationship satisfaction, or relationship commitment”, obviously “cohabitators reported considerably higher companionship”. The study does not permit to know the potential differences between living or not living with romantic partner on relationship quality. Although the aim of your study was not to investigate relationship duration, attachment style and relationship quality of cohabiting couples of emerging adults, I think it is important make clear to readers the distinction between dating couples (non-cohabiting), cohabiting couples (not married), and married couples. Finally, if you exclude avoidant attachment style in the analysis, I think you should specifically mention “anxious attachment style” instead of “attachment insecurity” throughout the Discussion section.

Abstract

In the light of the above, if you do not include avoidant attachment style in the analysis, I think you should specifically refer to “anxious attachment style” instead of “attachment insecurity”, throughout the abstract. Moreover, I think you should specify the age range and the mean age of the sample of undergraduate students, as well as that, almost in its totality, it was composed by participants who were not living with their romantic partner.

Author Response

Thank you for your close reading of our manuscript and for your specific and constructive comments.  We revised and updated the article to provide more detail on our analysis.  We have, in fact, rerun our entire analysis to ensure that the avoidance items could not be incorporated.  We provide a point by point response to your feedback below. 

  1. Comment: Why did you not include avoidance subscale in the analysis?

Resolved: We originally planned to include the avoidant scale from the ECR in the analysis, however; a few of the items from the scale have incomplete data due to not being included in all semesters of data collection. Nonetheless, we entered three items from the ECR avoidant scale with complete data.  The model fit was compromised by cross-loadings with an insecurity parcel and with a relationship satisfaction parcel.  In addition, an avoidance item had lower than desirable standardized loading on the latent factor.  When we removed the avoidance items the model fit improved dramatically with no cross-loadings.  Although we discussed this in the original article, we did not provide any specific information on model fit indices.  We have now added a detailed footnote to the measurement model and a reference to this footnote in the measurement section.  The footnote reads:

“The  modification indices suggested potential cross loadings of an avoidance item (not showing deep feelings) on the insecurity factor as well as a cross-loading of a relationship satisfaction indicator (LSI expectation) on the avoidance factor. In addition, one of the avoidance items had a somewhat low standardized loading of 0.57. The model with anxious attachment items and not avoidance items had better fit indices and a clearer factor structure.”

  1. Comment: if you do not include avoidant attachment style in the analysis, I think you should specifically refer to “anxious attachment style” instead of “attachment insecurity”, throughout the abstract., if you exclude avoidant attachment style in the analysis, I think you should specifically mention “anxious attachment style” instead of “attachment insecurity” throughout the Discussion section.

 Resolved: We updated the abstract and discussion to reflect our focus on anxious attachment only

  1. Comment: Please, add the Hadden’s results referred to avoidant attachment.

Resolved: We updated our reference to the Hadden et al., study to include their findings on avoidant attachment.

  1. Comment: Add to limitations: the results are not generalizable to emerging adulthood (18-29 years) because participants were undergraduate students and ranged in age from 18 to 23 years (M=19.7, SD=1.77). I think you should specify the age range and the mean age of the sample of undergraduate students, as well as that, almost in its totality, it was composed by participants who were not living with their romantic partner.

Resolved: To make clear the demographic limitations of our sample (i.e., non-cohabitating, college attending, and early emerging adulthood).  We added to our discussion the need to study dating relationship patterns among an older community-based sample given that unmarried older adults in career positions are likely to approach dating with different expectations and, therefore, may show different relationship trajectories. 

Reviewer 3 Report

What a pleasure to read!

That is my overall impression after reading this manuscript.  The paper is written beautifully for the most part, and the argument is generally compelling.  Here are some thoughts I’d like to share.

First, we frequently hear/read about, for instance, Gen Z’s “messed up” value priorities when it comes to dating, marriage, childbearing, etc., yet there hasn’t been much scholarly research on the topic.  FOR SURE, there is a gap in the literature, and I particularly appreciate the part where the authors articulate that we in the research world are STILL stuck with the traditional presumption of dating being a precursor to marriage, after which heterosexual couples are to have biological children.  Very nice!

At the same time, though, the abstract does not appear to do this paper justice because it does not really “sell” this very important aspect of the work being reported here.  I do believe this is a sensational focus—that our old presuppositions around dating may not hold true in today’s society where young people are doing things “out of order” from our perspective, which makes the traditional views possibly obsolete in many ways.  The abstract, as we have it, seems to focus on factual details without the big picture around the major contributions this work makes.  So, although the abstract alone will not make or break this submission, I’d like to note that the abstract could be edited to package this manuscript in a far more appealing way (especially for people that are merely browsing through the abstracts).

Now, while the argument is generally well-supported by references, the references DO feel a bit dated.  I wonder if this paper has been in “incubation” for a while—or maybe the authors simply didn’t find very many relevant references from the last decade or so.  With so many papers out on romantic relationship quality over the last few years alone (including the Eastwick et al. reference from 2019, which is prominently citated here), I somewhat suspect the former to be more of a case than the latter.  I am sure the readers of this paper, too, will have similar impressions, so I’d love to invite the authors to consider updating the references a bit more.

I do understand that there might not be references directly targeting the relationship quality over time, but more recent works will, for sure, serve as great citations for the peripheral points being made—such as the life-course approach, such as this one rather than/in addition to those from 20+ years ago: Gere, & Impett, E. A. (2018). Shifting priorities: Effects of partners’ goal conflict on goal adjustment processes and relationship quality in developing romantic relationships. Journal of Social and Personal Relationships, 35(6), 793–810. https://doi.org/10.1177/0265407517698851. 

In discussing limitations, the authors allude to, among others, SES.  There is an interesting paper on relationship quality and equity/power differentials, which should really be included as a reference in addition to those from 10 to 30 years ago to illustrate the future need to address this issue in today’s DEI-centric society:

Cho, M., Impett, E. A., Campos, B., Chen, S., & Keltner, D. (2020). Socioeconomic inequality undermines relationship quality in romantic relationships. Journal of Social and Personal Relationships, 37(5), 1722–1742. https://doi.org/10.1177/0265407520907969

The results are presented in an orderly manner, and the discussions follow the results nicely.

Overall, again, I do believe this paper has been beautifully written, yet it somehow feels a bit dated in some places.  Updating the references should definitely help, and I do not anticipate this to be very difficult. 

Thank you so much for the opportunity to review this delightful manuscript, and I look forward to seeing it in print with minor revisions.

Author Response

First, we want to thank the reviewer for their gracious and affirming feedback – you did make these researchers feel appreciated! 

We detail our response to your comments below. 

  1. Comment: The abstract does not appear to do this paper justice because it does not really “sell” this very important aspect of the work being reported here. The abstract, as we have it, seems to focus on factual details without the big picture around the major contributions this work makes.  

Resolved: Thank you for this comment.  We revised the intro to the abstract to convey more of the study’s importance and implications.  The first three sentences now read:

“For many young adults today dating is not taken as a path to marriage, but as a relationship to be considered on its own terms with a beginning, middle, and end. Yet, research has not kept pace as most studies that look at relationships over time focus on marriages. In the present study, we look at individual differences and normative patterns of dating relationship quality over time. 

R Comment. I invite the authors to consider updating the references a bit more.

Resolved: You are absolutely correct that the references in this paper have sat for a few years.  Our updates have focused on research directly examining relationship quality over time, and there are no new references that we could locate for this focal area.  However, we located several recent studies that provide background and peripheral findings that support our study and provide springboards for continue work that includes duration and dating quality as factors. We added seven new citations to the introduction and five to the discussion, including the two recommended by the reviewer. The 12 added studies and reviews were published in the previous 3 years.

Beeney, J. E., Stepp, S. D., Hallquist, M. N., Ringwald, W. R., Wright, A. G. C., Lazarus, S. A., Scott, L. N., Mattia, A. A., Ayars, H. E., Gebreselassie, S. H., & Pilkonis, P. A. (2019). Attachment styles, social behavior, and personality functioning in romantic relationships. Personality Disorders: Theory, Research, and Treatment, 10(3), 275–285. https://doi.org/10.1037/PER0000317

Bühler, J. L., Krauss, S., & Orth, U. (2021). Development of relationship satisfaction across the life span: A systematic review and meta-analysis. Psychological Bulletin, 147(10), 1012. https://doi.org/10.1037/BUL0000342

Cho, M., Impett, E. A., Campos, B., Chen, S., & Keltner, D. (2020). Socioeconomic inequality undermines relationship quality in romantic relationships. Https://Doi.Org/10.1177/0265407520907969, 37(5), 1722–1742. https://doi.org/10.1177/0265407520907969

Christensen, M. A. (2020). “Tindersluts” and “Tinderellas”: Examining the Digital Affordances Shaping the (Hetero)Sexual Scripts of Young Womxn on Tinder. Https://Doi.Org/10.1177/0731121420950756, 64(3), 432–449. https://doi.org/10.1177/0731121420950756

Díez, M., Sánchez-Queija, I., & Parra, Á. (2019). Why are undergraduate emerging adults anxious and avoidant in their romantic relationships? The role of family relationships. PLOS ONE, 14(11), e0224159. https://doi.org/10.1371/JOURNAL.PONE.0224159

Feeney, J., & Fitzgerald, J. (2019). Attachment, conflict and relationship quality: laboratory-based and clinical insights. Current Opinion in Psychology, 25, 127–131. https://doi.org/10.1016/J.COPSYC.2018.04.002

Gere, J., & Impett, E. A. (2017). Shifting priorities. Https://Doi.Org/10.1177/0265407517698851, 35(6), 793–810. https://doi.org/10.1177/0265407517698851

Karney, B. R., & Bradbury, T. N. (2020). Research on Marital Satisfaction and Stability in the 2010s: Challenging Conventional Wisdom. Journal of Marriage and Family, 82(1), 100–116. https://doi.org/https://doi.org/10.1111/jomf.12635

Kayabol, N. B. A., Gonzalez, J. M., Gamble, H., Totenhagen, C. J., & Curran, M. A. (2020). Levels and volatility in daily relationship quality: Roles of daily sacrifice motives. Https://Doi.Org/10.1177/0265407520945032, 37(12), 2967–2986. https://doi.org/10.1177/0265407520945032

Olmstead, S. B. (2020). A Decade Review of Sex and Partnering in Adolescence and Young Adulthood. Journal of Marriage and Family, 82(2), 769–795. https://doi.org/10.1111/JOMF.12670

Ribeiro, F. N., Sousa-Gomes, V., Moreira, D., Moreira, D. S., Oliveira, S., & Fávero, M. (2022). The Relationship Between Romantic Attachment, Intimacy, and Dyadic Adjustment for Female Sexual Function. Sexuality Research and Social Policy 2022 19:4, 19(4), 1920–1934. https://doi.org/10.1007/S13178-022-00738-3

Sheng, R., Hu, J., Liu, X., & Xu, W. (2022). Longitudinal relationships between insecure attachment and romantic relationship quality and stability in emerging adults: the mediating role of perceived conflict in daily life. Current Psychology 2021, 1–11. https://doi.org/10.1007/S12144-021-02668-6